# Comparative Study of the Myocardium of Patients from Four COVID-19 Waves

**DOI:** 10.3390/diagnostics13091645

**Published:** 2023-05-07

**Authors:** Lubov Borisovna Mitrofanova, Igor Aleksandrovich Makarov, Andrey Nikolaevich Gorshkov, Andrey Leonidovich Runov, Maxim Sergeevich Vonsky, Maria Mikhailovna Pisareva, Andrey Borisovich Komissarov, Taiana Alekseevna Makarova, Qingli Li, Tatiana Leonidovna Karonova, Alexandra Olegovna Konradi, Evgeny Vladimirovich Shlaykhto

**Affiliations:** 1Almazov National Medical Research Centre, St. Petersburg 197341, Russia; mitrofanova_lb@almazovcentre.ru (L.B.M.);; 2Smorodintsev Research Institute of Influenza, St. Petersburg 197376, Russia; 3D.I. Mendeleyev Institute for Metrology, St. Petersburg 190005, Russia; 4East China Normal University, Shanghai 200241, China

**Keywords:** COVID-19 waves, myocarditis, immunohistochemical, immunofluorescent myocardial study, ultrastructural myocardial study, polymerase chain reaction

## Abstract

Background: Few studies have compared COVID-19 patients from different waves. This study aims to conduct a clinical and morphological analysis of patients who died from COVID-19 during four waves. Methods: The study involved 276 patients who died from COVID-19 during four waves, including 77 patients in the first wave, 119 patients in the second wave, and 78 patients in the third wave. We performed a histological examination of myocardium samples from autopsies and additionally analyzed the samples by PCR. We conducted immunohistochemistry of the myocardium for 21 samples using antibodies against CD3, CD45, CD8, CD68, CD34, Ang1, VWF, VEGF, HLA-DR, MHC1, C1q, enteroviral VP1, and SARS-CoV-2 spike protein. We also did immunofluorescent staining of three myocardial specimens using VP1/SARS-CoV-2 antibody cocktails. Further, we ran RT-ddPCR analysis for 14 RNA samples extracted from paraffin-embedded myocardium. Electron microscopic studies of the myocardium were also performed for two samples from the fourth wave. Results: Among the 276 cases, active myocarditis was diagnosed in 5% (15/276). Of these cases, 86% of samples expressed VP1, and individual cells contained SARS-CoV-2 spike protein in 22%. Immunofluorescence confirmed the co-localization of VP1 and SARS-CoV-2 spike proteins. ddPCR did not confidently detect SARS-CoV-2 RNA in the myocardium in any myocarditis cases. However, the myocardium sample from wave IV detected a sub-threshold signal of SARS-CoV-2 by qPCR, but myocarditis in this patient was not confirmed. Electron microscopy showed several single particles similar to SARS-CoV-2 virions on the surface of the endothelium of myocardial vessels. A comparison of the cardiovascular complication incidence between three waves revealed that the incidence of hemorrhage (48 vs. 24 vs. 17%), myocardial necrosis (18 vs. 11 vs. 4%), blood clots in the intramural arteries (12 vs. 7 vs. 0%), and myocarditis (19 vs. 1 vs. 6%) decreased over time, and CD8-T-killers appeared. Immunohistochemistry confirmed the presence of endotheliitis in all 21 studied cases. Conclusions: This study compared myocardial damage in patients who died during three COVID-19 waves and showed a decrease in the incidence of endotheliitis complications (thrombosis, hemorrhage, necrosis) and myocarditis over time. However, the connection between myocarditis and SARS-CoV-2 infection remains unproven.

## 1. Introduction

Many authors have shown a milder course of the disease and lower mortality among hospitalized patients with COVID-19 during the second wave in comparison with the first wave [1,2]. However, only a few studies have performed a comprehensive comparison of clinical characteristics, including laboratory results, between the patients from different waves [3,4,5]. Even less is known about the comparative pathomorphological data of the patients.

The high incidence of myocardial injury among hospitalized patients with COVID-19 (from 13 to 41% respectively) leads to increased mortality from cardiovascular disease [6]. The prevalence of myocarditis among COVID-19 patients remains challenging to estimate due to the lack of adequate diagnostic because of the hospital services overload [7]. According to a single publication, 5% of patients manifested new acute myocarditis [8]. At the same time, another reported less than a 2% incidence of myocarditis [9,10]. Both in direct viral infection of the myocardium and indirect injury through inflammation, endothelial activation and microvascular thrombosis are considered to occur in the progression of COVID-19. Cardiovascular damage is determined by the effect of the SARS-CoV-2 binding to the angiotensin-converting enzyme 2 (ACE2) receptor in the renin-angiotensin system, the body’s innate immune response, and the vascular response to cytokine production [11]. The extent of the cardiovascular injury is affected by the amount of viral inoculum, the immune response magnitude, and comorbidities. Due to the ubiquitous distribution of SARS-CoV-2 receptor ACE2, a large number of organs become potential targets for infection, e.g., heart, liver, brain, pancreas, and kidneys [12].

Despite numerous studies, it is still unclear whether SARS-CoV-2 causes myocarditis, as many researchers did not detect SARS-CoV-2 genomes using RT-qPCR in the myocardium of the patients [13,14] or detected the presence of SARS-CoV-2 in patients without myocarditis [15,16]. In addition, ultrastructural studies provided controversial results. In particular, Tavazzi G et al. [17] demonstrated the presence of the virus in myocardial macrophages, Fox SE et al. [18] showed the virus localization in endotheliocytes, while Bulfamante GP et al. [19] analyzed autopsies of patients without clinical signs of heart disease but found the virus in cardiomyocytes. At the same time, reliable detection of SARS-CoV-2 virions using electron microscopy (EM) is challenged due to the structural similarity between virions and intracellular components, causing a lack of agreement in the literature [20].

It is important to note that the results of many studies did not exclude co-infection of the myocardium with SARS-CoV-2 and another virus (REF).

The present study aimed to compare clinical traits and pathomorphological characteristics of myocardium between patients deceased during four COVID-19 waves.

## 2. Methods

### 2.1. Patients

The study included 276 patients deceased during four COVID-19 waves. Case histories of all patients included in the study were analyzed. The presence of SARS-CoV-2 was confirmed for all patients by RT-qPCR of antemortem nasopharyngeal swabs and autopsy materials, including tracheal, bronchial, and lung tissues. The RT-qPCR was performed using the SARS-CoV-2/SARS-CoV Reagent Kit (DNA-Technology LLC, Moscow, Russia, Lot F2912S-2M) on a CFX96 Touch amplifier. According to the procedure for handling death cases during the COVID-19 pandemic implemented in Russia in March 2020, case histories, autopsy protocols, and histological preparations of all deceased patients with COVID-19 were submitted for a systematic, centralized evaluation by an expert commission appointed by the Saint Petersburg Committee for Health Protection to determine the cause of death and its connection with COVID-19 infection [21]. For 270 patients (98%), a computed tomography scan (Somatom Definition 128 Siemens) was performed at admission, along with clinical blood testing of C-reactive protein, procalcitonin, D-dimer, ferritin, and troponin I measurements. In addition, most patients underwent electrocardiographic (ECG) and echocardiographic studies.

Patients from Wave I. The cohort included 77 patients who died from acute respiratory distress syndrome secondary to bilateral viral bronchopneumonia during a period of 1 to 57 days after hospitalization in April–July 2020. Among them were 34 women and 43 men aged 34 to 94 years, and the median age was 71 years (IQR 66–76). The median time from symptom onset to admission was 4 days (IQR 3–6) and the median time between the onset of symptoms and death was 18 days (IQR 14–20). The median number of bed-days was 14 (IQR 11–16). All sequenced samples from this period were attributed to the B.1 lineage and its sublineages of COVID-19 [22]. All patients in this cohort did not receive COVID-19 vaccination.

Patients from Wave II. The cohort included 119 patients who died from acute respiratory distress syndrome in December 2020–February 2021. Among them were 63 women and 56 men aged 40 to 98 years, and the median age was 74 years (IQR 71–80). The median time from symptom onset to admission was 7 days (IQR 6–7), and the median time between the onset of symptoms and death was 19 days (IQR 17–22). The median number of bed-days was 12 (IQR 11–15). In Saint Petersburg, two Russian endemic lineages of SARS-CoV-2 were detected in that period, the B.1.397 and B.1.317. The Alpha VOC (B.1.1.7) and another lineage, AT.1, which has probably emerged in St. Petersburg, were first detected in February 2021 [21]. All patients in this cohort did not receive COVID-19 vaccination.

Patients from Wave III. The cohort included 78 patients who died during the third wave of COVID-19 in June–August 2021. Among them were 48 women and 30 men aged 29 to 93 years, and the median age was 72 years (IQR 68–76). The median time from symptom onset to admission was 8 days (IQR 7–9) and the median time between the onset of symptoms and death was 21 days (IQR 16–24). The median number of bed-days was 13 (IQR 9–15). In Saint Petersburg, only the B.1.617.2 lineage (the Delta VOC) of SARS-CoV-2 was detected in that period. Four patients received one shot of Gam-COVID-Vac vaccine (Sputnik V), and three patients were fully vaccinated with two doses with a three-week interval.

Patients from Wave IV. From the fourth wave, we analyzed autopsy samples from two patients (an 88-year-old woman and a 79-year-old man), who died from acute respiratory distress syndrome in March 2022 (on the 7th and 45th day of hospitalization, respectively). In Saint Petersburg, only the B.1.1.529 lineage (Omicron) of SARS-CoV-2 was detected in that period. No patients in this cohort received COVID-19 vaccination. We were only able to include two patients with COVID-19 caused by Omicron, as these patients were the only ones who matched the severity of infection seen in the other participants, meeting the criteria of death from acute respiratory distress syndrome secondary to bilateral viral bronchopneumonia and confirmed presence of SARS-CoV-2 in antemortem nasopharyngeal swabs and autopsy materials, including tracheal, bronchial, and lung tissues (RT-qPCR).

COVID-19 was confirmed for all patients by RT-PCR of antemortem nasopharyngeal swabs and autopsy materials including tracheal, bronchial, and lung tissues. The RT-qPCR was performed using commercial kits registered in the Russian Federaton. According to the procedure for handling death cases during the COVID-19 pandemic implemented in Russia in March 2020, case histories, autopsy protocols, and histological preparations of all deaths with COVID-19 were submitted for a systematic centralized evaluation by an expert commission appointed by the St. Petersburg Committee for Health Protection to determine the cause of death and its possible connection with COVID-19 infection [21].

For the majority of patients (98%), a CT scan was performed (on a Somatom Defenition 128 Siemens tomograph) and standard laboratory examination with evaluation of a clinical blood test for C-reactive protein, procalcitonin, D-dimer, ferritin, and troponin I at admission. In addition, most patients underwent electrocardiographic (ECG) and echocardiographic studies.

### 2.2. Histological Examination

Myocardium samples were taken from ventricular and atrial free walls and the interventricular septum. The routine histological examination of the myocardium was carried out with hematoxylin and eosin staining in all 276 patients.

### 2.3. Immunohistochemical Study

Immunohistochemical (IHC) analysis using specific antibodies was performed for specimens with non-purulent inflammatory myocardial infiltration. IHC analysis of myocardium samples (15 patients from wave I, 1 patient from wave II, 5 patients from wave III) was carried out according to a standard protocol [23] using antibodies against the following targets: CD3 (rabbit polyclonal antibody, DAKO, Nottingham, UK; dilution 1:100), CD45 (mouse monoclonal antibody, clone 2B11e+PD7/26; DAKO, Nottingham, UK; dilution 1:400), CD8 (mouse monoclonal antibody, clone 1A5; Leica Biosystems, UK; dilution 1:20), CD68 (mouse monoclonal antibody, clone PG-M1; DAKO, Carpinteria, CA, USA; dilution 1:25), CD34 (mouse monoclonal antibody, clone QBEnd/10; DAKO, UK; dilution 1:50), Ang1 (rabbit polyclonal antibody, Abnova, Walnut, CA, USA; dilution 1:50), VWF (mouse monoclonal antibody, clone F8/86; RTU; Diagnostic BioSystems, Slough, UK), VEGF (mouse monoclonal antibody, clone VG1; Diagnostic BioSystems, Slough, UK; dilution 1:50), HLA-DR (mouse monoclonal antibody, clone LN3; Leica, USA; dilution 1:150), MHC1 (mouse monoclonal antibody, clone EMR8-5; Abcam, Waltham, MA, USA; dilution 1:1000), C1q (rabbit polyclonal antibody, Diagnostic BioSystems, Slough, UK, dilution 1:100), enteroviral VP1 (mouse monoclonal antibody, 5-D8/1; DAKO; UK dilution 1:25), and SARS-CoV-2 Spike protein (rabbit polyclonal antibody; GeneTex, Hsinchu City, Taiwan; dilution 1:100). Micrographs were taken with a Leica DM4000 microscope.

### 2.4. Immunofluorescence Study

Immunofluorescence microscopy (IF) was performed on myocardium paraffin sections of three patients with myocarditis (where expression of enterovirus VP1 and SARS-CoV-2 Spike proteins was evidenced by IHC) with the following antibody cocktail: VP1/SARS-CoV-2 Spike protein (antibodies and dilutions as described above). Alexa Fluor 594 and Alexa Fluor 488 (Thermo Fisher Scientific, Waltham, MA, USA) were used as secondary antibodies. Sections were counterstained with DAPI (AppliChem, Gary, IN, USA). Micrographs were taken with a Leica DM6000B microscope.

### 2.5. SARS-CoV-2 RT-ddPCR

In the samples from patients with IHC-confirmed myocarditis (14 cases), SARS-CoV-2 genome RNA content was measured using droplet digital PCR after reverse transcription (RT-ddPCR, two-step) in accordance with the recommendations of CDC China (for gene N) and Charite research laboratory (for gene E) [24,25]. RNA was isolated from five 5-µm slices of formalin-fixed paraffin-embedded (FFPE) samples of left ventricular tissue blocks using AllPrep DNA/RNA FFPE Kit (QiaGen, Venlo, The Netherlands, 80234) according to the manufacturer instructions. Reverse transcription was performed using RevertAid H Minus First Strand cDNA Synthesis Kit (ThermoScientific, Waltham, MA, USA, K1632) with specific primers for genes N and E and random hexamer primers. Droplet digital PCR was run on the Bio-rad QX200 Droplet Digital System with ddPCR Supermix for Probes (Bio-Rad, Hercules, CA, USA, #1863026). Primers and probe sequences are presented in Table 1. The quality of isolated RNA was confirmed by RT-ddPCR analysis of β-actin mRNA target. The sensitivity of the assay was assessed using Bio-Rad SARS-CoV-2 Standard reference material sequential dilutions.

For the myocardium autopsy specimens from two patients who died during the fourth wave, SARS-CoV-2 RT-qPCR was performed using the commercial kit “Intifica SARS-CoV-2” (Alkor Bio Group, Saint Petersburg, Russia).

## 3. Morphometry and Statistics

The morphometric analysis included calculation of the average number of inflammatory infiltrate cells per 1 mm^2^ and assessment of the expression of HLA-DR, MHC1, C1q, Ang1, VWF, VEGF, enteroviral VP1, and SARS-CoV-2 Spike protein. Results were scored as follows: 1—expression on single cells and vessels, 2—on 1/3 of the cells, 3—on 1/2–2/3 of the area of the preparation, and 4—on all cells under study. Morphometric analysis was performed using an automated image analyzer (Image Scope Color M, Saint Petersburg, Russia).

Myocardial inflammation was diagnosed by the presence of ≥14 leucocytes/mm^2^ including up to 4 monocytes/mm^2^, with the presence of CD3-positive T lymphocytes ≥ 7 cells/mm^2^ according to the European Society of Cardiology (ESC) position [26].

Statistical analysis of the acquired data was performed using IBM SPSS Statistics 28.0.1.0. All distributions analyzed differed from normal. Therefore, nonparametric methods of analysis were used, including the analysis of variance, the Mann–Whitney test for independent samples, and the Wilcoxon test. For nominal rank variables, we built cross-tabulations and evaluated Pearson’s chi-square test. Differences between groups were defined as significant at *p* < 0.05. Primary data and their statistical analysis are described in Appendix A.

### Electron Microscopy

The myocardium samples of two patients from the fourth wave were examined via electron microscopy. The samples were collected within the first 6 h after death. Each specimen was cut into small pieces, approximately 1–2 mm^3^ in size, and pre-fixed with 2.5% glutaraldehyde in PBS (pH 7.4) for 45 min at room temperature. These pieces were washed three times with PBS and post-fixed in 1% PBS-buffered OsO_4_ for 1 h. Specimens were then dehydrated in a series of ethanol solutions of gradually increasing concentration and embedded in Epon epoxy resin. Ultra-thin sections (70–90 nm) were obtained using a Leica EM UC7 ultramicrotome. Sections were transferred onto copper grids, stained with uranyl acetate and lead citrate, and examined using an HT7800 transmission electron microscope (Hitachi, Tokyo, Japan).

## 4. Results

### 4.1. Clinical and Morphological Characteristics of the Myocardium of Patients from the First Wave

Of the 85 patients, anamnestic data showed the presence of arterial hypertension (AH) in 49 (64%), ischemic heart disease (IHD) in 42 (55%), diabetes mellitus (DM) in 15 (20%), chronic cerebrovascular disease (CCVD) with acute cerebrovascular accident in 2 (3%), oncological diseases in 10 (13%), chronic obstructive pulmonary disease (COPD) in 3 (4%), and HIV infection in 1 (1%). Levels of C-reactive protein, D-dimer, ferritin, and procalcitonin were elevated in all patients. The median percentage of lung parenchyma lesions on CT upon admission to the hospital was 76% (IQR 60–80%). Most patients had lymphopenia (Table 2).

Analysis of case histories of patients revealed active lymphocytic myocarditis in 9 out of 15 cases, rhythm and conduction disturbances presented with atrial fibrillation in 4 cases, atrial extrasystole in 1 case, ventricular extrasystole in 1 case, incomplete right bundle branch block in 4 cases, and complete left bundle branch block in 2 cases. For one patient, an ECG was not performed. ICH confirmed active lymphocytic myocarditis in accordance with international criteria [8] only in 8 out of 15 cases (53%), which corresponded to 10% of the total number of deceased patients from the first wave (Table 3). In 75% (6/8) of patients with IHC-confirmed myocarditis, an increase in troponin I level was detected. However, of clinical and laboratory data analysis did not find statistically significant differences between patients with and without myocarditis (Table 2).

IHC study of eight active lymphocytic myocarditis from the first wave revealed expression of MHC1 in all samples, HLA-DR in five samples, expression of the C1q component of the complement in six samples, and presence of capsid VP1 antigen of enteroviruses mainly in the walls of vessels in six samples. Expression of CD34 was determined on large polymorphic cells in the vascular zone (Figure 1). Only a fraction of these cells were macrophages expressing CD68. An increased number of macrophages (≥14/mm^2^) was determined in three out of eight cases. Histological and immunohistochemical studies did not display a pronounced diffuse lymphocytic infiltration of the myocardium. The maximum focal infiltrates were 21–37 CD3+T-lymphocytes/mm^2^. CD8+T-lymphocytes were not found in any case of myocarditis. In all eight cases of lymphocytic myocarditis, expression of von Willebrand factor (VWF) was observed on the vascular endothelium, while the expression of angiopoietin Ang1 was found on the endothelium and in the sarcoplasm of cardiomyocytes. Expression of the vascular growth factor VEGF on the endothelium and in muscle fibers was observed in four cases, only on the endothelium in three cases, and was absent in one case. Expression of SARS-CoV-2 Spike protein was detected only in one case in single endothelial cells. To sum up, we found that the expression of Ang1, VEGF, and VWF was regardless of the presence or absence of active lymphocytic myocarditis in 15 patients deceased during the first wave of COVID-19. Moreover, Ang1 and VEGF were expressed not only in the endothelium, but also in cardiomyocytes in some cases.

RT-ddPCR did not confidently confirm SARS-CoV-2 RNA presence in FFPE tissue specimens from the patients with myocarditis.

### 4.2. Clinical and Morphological Characteristics of the Myocardium of Patients from the Second Wave

Among 119 patients deceased during the second wave, AH was documented in 106 (89%), IHD was recorded in 85 (72%), including 7 patients with coronavirus infection proceeding on the background of acute myocardial infarction (AMI), type 2 DM was registered in 54 (46%), 1 (1%) had CCVD with a history of acute cerebrovascular accident, 15 (12%) had oncological diseases, and 10 (9%) had COPD. In addition, one patient had a history of HIV infection. Heart rhythm and conduction disturbances were also recorded in most patients. AF was observed in 51 (43%) deceased patients, bundle branch block in 29 (25%), AV block in 11 (9%), sick sinus syndrome in 4 (3%), ventricular extrasystoles in 4 (3%), and atrial extrasystoles in 4 (3%).

Myocarditis was suspected (results of the histological examination are presented in Table 4) and confirmed by IHC in only one case (less than 1%). This patient had active lymphocytic myocarditis with a pronounced inflammatory reaction in the myocardium, which we have not observed in other waves.

### 4.3. Clinical and Morphological Characteristics of the Myocardium of Patients from the Third Wave

Among 78 patients who died during the third wave, AH was documented in 73 (94%), IHD was recorded in 32 (41%), including 1 patient with AMI, type 2 DM was registered in 38 (49%), 24 (31%) had CCVD chronic cerebrovascular disease with a history of acute cerebrovascular accident, 7 (9%) had oncological diseases, and 4 (5%) had COPD. Heart rhythm and conduction disturbances were also recorded for most patients, thus, AF was observed in 30 (39%) deceased patients, bundle branch block in 33 (26%), AV block in 2 (3%), and ventricular extrasystoles in 6 (8%).

IHC-confirmed lymphocytic active myocarditis was detected in five cases (7%). In one case, myocarditis was diagnosed in a patient who developed disease symptoms a week after the first shot of Sputnik V (Table 4, line 11). CD8+T-killers were detected in two cases (Figure 2).

IHC analysis of all myocarditis cases from wave II and III revealed the expression of MHC1, VWF, Ang1, VEGF, and enteroviral VP1 (mainly in the walls of blood vessels; Table 4, lines 9–14); in five out of six cases—HLA-DR, in two out of six cases—CD8, and in four out of six cases—an expression of the C1q component of the complement. Expression of SARS-CoV-2 Spike protein was detected in single endotheliocytes and macrophages and in cardiomyocytes in two cases (Figure 2 and Figure 3).

Only single unreproducible SARS-CoV-2-positive droplets were identified by RT-ddPCR of RNA isolated from FFPE samples of myocardium tissues from patients with myocarditis from the second and third waves (Figure 4). The appearance of these droplets did not allow us to confirm the presence of viral RNA in the analyzed samples confidently.

Enterovirus VP1 protein and SARS-CoV-2 Spike protein were found to be colocalized in the endothelium and cardiomyocytes in one specimen from the first wave and two specimens from the third wave, as was revealed by IF (Figure 5).

### 4.4. Clinical and Morphological Characteristics of the Myocardium of Patients from the Fourth Wave

An 88-year-old patient and a 79-year-old patient were diagnosed with AH and IHD. Histological examination did not reveal acute pathological changes in the myocardium in both cases. Nevertheless, the myocardium autopsy specimen from the first patient was positive for SARS-CoV-2 in RT-qPCR for three gene fragments (N, ORF1, and ORF8; Table 4).

**Table 4 diagnostics-13-01645-t004:** Results of RT-qPCR for SARS-CoV-2 for two patients from the fourth wave.

Myocardial Sample	Cq ORF1	Cq ORF8	Cq N
№ 1	31, 10	32, 85	34, 27
№ 2	NEGATIVE	NEGATIVE	NEGATIVE

The RT-qPCR data were confirmed by the EM analysis of the myocardial autopsy of this patient. On the apical surface of some endotheliocytes, we found few single rounded membrane particles with a diameter of 75–120 nm, which have a clear morphological similarity to SARS-CoV-2 virions (Figure 6). By itself, EM does not allow identification of SARS-CoV-2 virions reliably on ultrathin autopsy sections since other cellular structures, such as transverse sections of microvilli, can be interpreted mistakenly as virions. However, in combination with RT-qPCR data (Table 5) and immunohistochemical detection of SARS-CoV-2 Spike protein in the endothelium of myocardial vessels (Figure 4), we can attribute these particles on electron micrographs as SARS-CoV-2 virions with a high level of confidence. It is also worth mentioning that EM analysis of the myocardial autopsy of the second patient (with a negative RT-qPCR result for SARS-CoV-2) did not reveal such particles in the endothelium.

In summary, active myocarditis were detected in 5% (14/276) of patients who had undergone hospitalization due to coronaviral infection during the four waves. IHC analysis of patients with myocarditis showed the expression of the enterovirus VP1 protein in the myocardium of 86% (12/14) of patients, and SARS-CoV-2 Spike protein expression in single cells in 22% (3/14) of patients. RT-qPCR analysis did not detect SARS-CoV-2 RNA in the myocardium in any of the myocarditis cases of wave I–III. At the same time, RT-qPCR in one of two cases in wave IV revealed a slightly positive reaction to SARS-CoV-2, but myocarditis in this patient was not confirmed morphologically. EM has demonstrated single structures similar to viral particles.

Myocardial vessel disorders were similar in all patients of the three waves (Table 5). They were manifested (summary in 276 patients) either by the destruction of the wall with hemorrhages in 28% of cases or by thrombosis in 7% of cases with myocardial necrosis. Focal plasma impregnation of the intima of the intramural arteries was determined in 15% of cases, and infiltrative vasculitis was found in 3%. Hypertrophied polymorphic endothelial cells with dysplasia were found to a greater extent, 56% (43/77), of the samples from the first wave (Figure 7). In an IHC, endothelial activation was confirmed by the expression of MHC1, VWF, Ang1, and VEGF in all samples.

We did not find significant statistical differences in clinical and laboratory parameters (see Table 1) between the patients with and without myocarditis (Figure 8).

Comparative analysis of patients from the first wave of COVID-19 and patients from the second and third waves showed that the proportion of patients with myocarditis decreased (*p* < 0.05). The count of macrophages in the myocardial stroma was the highest in the second wave (*p* = 0.017). It decreased in wave III compared to waves I and II (*p* = 0.008). The number of hypertrophied endotheliocytes with dysplasia dramatically decreased (*p* = 4.96 × 10^−14^); however, the number of cases with focal plasma impregnation of the intima of the intramural arteries increased (*p* = 0.0064) in the myocardium of patients who died during wave III. By wave III, the number of non-coronary ischemic necrosis (*p* = 0.004) and thrombosis (*p* = 0.013) patients decreased. At the same time, the immunohistochemical signs of endothelial overactivation persisted in patients with myocarditis.

In the first wave of COVID-19, increased levels of hs-cTnI were observed in 27 pa-tients (35% of the total). Of these, only 8 (30%) patients had elevated levels associated with acute myocardial infarction, 5 (22%) with myocarditis, and the remaining 13 (48%) patients had morphological manifestations of endotheliitis and its complications, such as thrombosis, bleeding, and necrosis.

In the second wave, increased levels of hs-cTnI were observed in 17 patients (14% of the total). Of these, only 6 (35%) patients had elevated levels associated with acute myocardial infarction, while the remaining 11 (65%) patients had morphological manifestations of endotheliitis, focal plasma impregnation, and fragmentation of muscle fibers. None of the patients with elevated levels of hs-cTnI were histologically verified with myocarditis.

In the third wave, increased levels of hs-cTnI were observed in 11 patients (14% of the total). Of these, only one (9%) patient had elevated levels associated with acute myo-cardial infarction, two (18%) with myocarditis, and the remaining eight (73%) patients had morphological manifestations of endotheliitis, plasma impregnation, interstitial edema of the myocardium, and fragmentation of muscle fibers.

Thus, in total for the three waves, we observed an increase in troponin in only 8 out of 14 patients (57%) with myocarditis. We can explain these findings by the possibility that patients in whom we did not observe an increase in the level of cardiac enzymes had minimal myocarditis activity and mild myocytolysis.

Rhythm and conduction disturbances were recorded in 62% (5/8) of patients with IHC-confirmed myocarditis from the first wave and in 60% (3/5) of patients from the third wave. Statistically significant correlations between the frequency and nature of arrhythmias in all waves were not found. There were also no correlations for the presence/absence of arrhythmias between patients with and without myocarditis. For more information on the clinical data of patients with myocarditis from the first, second, and third waves, see Appendix A.

## 5. Discussion

A comparison of patients of the three waves showed that myocardial damage during wave I was significantly more pronounced, which complies with numerous reports on distinct severity of clinical cases from the first wave [27,28]. No significant differences in the clinical and laboratory data of patients with and without myocarditis were observed in our study. The spectrum of concomitant diseases remained the same, which is consistent with the results of other studies [29,30]. Thus, according to our data, there are no clinical and laboratory predictors of myocarditis development on the background of COVID-19.

In our opinion, the main changes in the myocardium were caused by the so-called dysendotheliosis or vascular endotheliitis/endotheliopathy (hypertrophy, hyperplasia, dysplasia, or damage of the endothelium) with subsequent hemorrhages and/or thrombosis. Varga Z et al. [31] also described signs of endotheliitis in the lungs, heart, kidneys, and intestines, considering it to be a direct consequence of viral damage and inflammatory response in the host organism. According to the authors, COVID-19 endotheliitis could explain systemic microcirculatory disorders that occur in various organs.

In the present study, we demonstrated a significant decrease in the morphological manifestations of the so-called endotheliitis or dysendotheliosis and its consequences in the form of necrosis and hemorrhage by the third wave. At the same time, the symptom “focal plasma soaking of the intima of the intramural arteries” came out on top. It is most likely that this symptom results from a prolonged spasm of the arteries. It could reflect both concomitant cardiovascular pathology and treatment consequences.

We detected morphological signs of myocarditis only in 5% of cases, which is consistent with the biopsy data reported by other authors [32,33]. At the same time, there is no evidence that myocarditis was caused by SARS-CoV-2 infection since RT-ddPCR did not detect SARS-CoV-2 RNA in the myocardium, and IHC revealed SARS-CoV-2 Spike protein only in single cells in 3 out of 14 cases. In our opinion, currently, there is no clear evidence of coronavirus-associated myocarditis. This conclusion is in agreement with other researchers’ opinions [33,34]. It is most likely that the coronavirus infection triggers the activation of other viruses, such as enteroviruses. In line with that, in 72% of cases, we saw the expression of enteroviruses VP1 capsid in the myocardium, which predominated in the vascular walls. Apparently, the SARS-CoV-2 can activate a latent enterovirus infection in patients with cardiovascular diseases, such as IHD [35], resulting from an impaired immune response. In addition, it has been proven that the endogenous proteases of coronaviruses and enteroviruses are similar in their structure and functions. These proteases process their viral polyproteins, and thus drugs binding to the active site could potentially target both viral groups [36].

Despite the fact that we rarely detected myocarditis in general, the number of such patients decreased in the second and third waves. However, immunohistochemical signs of endothelial activation remained, including increased expression of Ang1, VEGF, and VWF. VEGF and Ang1 are the two main angiogenic factors being investigated for the treatment of myocardial infarction. Cardio-specific hypoxia-induced co-expression of VEGF and Ang1 is known to improve perfusion and heart function by inducing angiogenesis and proliferation of cardiomyocytes, activating survival pathways, and reducing cell apoptosis [37]. Moreover, this hypoxia can be both a consequence of ischemia [38] and hypoxemic [39], which is fully consistent with the fact that the main target of coronavirus infection in the myocardium is the endothelium.

It is also known that mice exposed to hypoxia demonstrated an increase in VWF expression accompanied by the occurrence of thrombi in the heart and lungs [40], similar to COVID-19. VWF plays a key role in platelet adhesion and aggregation in coronary arteries with stenosis and unstable atherosclerotic plaques. Numerous studies have shown that VWF is a significant predictor of adverse cardiac events, including death. Its level increases during acute coronary syndrome and serves not only as a marker but also as an important effector in the pathogenesis of myocardial infarction. The central role of VWF in thrombosis has made it a promising target in research of new antithrombotic therapies [41].

The EM study of one specimen from the fourth wave without signs of myocarditis revealed the presence of single structures resembling viral particles on the surface of the endothelium. In this sample, RT-qPCR also confirmed a subthreshold signal for SARS-CoV-2. Analyzing our results of a morphological study, including IHC and EM data, we assumed that the SARS-CoV-2 causes the most significant damage to the endothelium, which was confirmed by other authors [42,43,44].

The present study showed that cardiac arrhythmias were associated not only with myocarditis. Several origins have been described: (1) hypokalemia or hypomagnesemia as a result of the disease itself (e.g., diarrhea), especially in the critically ill, or with certain treatments such as diuretics; (2) pharmacotherapeutic agents repurposed for the treatment of COVID-19 with the risk of QT interval prolongation and torsades de pointes; (3) due to the direct electrophysiological effect of cytokines on the myocardium [45]; and (4) rapid deterioration of kidney function and electrolyte disturbances [46]. Finally, endotheliitis has noticeable consequences, such as hemorrhages (in 43% of patients from the first wave), non-coronary myocardial necrosis (in 17% of patients from the first wave), hemorrhagic and ischemic myocardial infarctions (respectively in 6 and 13% of cases). All these changes could also lead to rhythm disturbances. In addition, patients with a coronavirus infection developed foci of fibrosis, which may also cause heart rhythm disturbances in the future.

Among all patients included in the study, only 1% (3/276) were fully vaccinated, and 1.4% (4/276) received only one shot of Sputnik V. Therefore, this work does not cover the effect of vaccination on the amelioration of myocardial injury resulting from SARS-CoV-2 viral infection of the myocardium. We concluded that the reduction of myocardial injury severity was associated with the appearance of new, less dangerous SARS-CoV-2 strains. This was also confirmed by Cascella et al. [47].

## 6. Conclusions

A comparison of myocardial lesions in patients who died during different waves of COVID-19 showed that although the virus was still detectable at subthreshold values in the myocardium of patients from the fourth wave, the occurrence of endotheliitis complications (thrombosis, hemorrhage, myocardial necrosis) and myocarditis significantly decreased. Nevertheless, the link between myocarditis and coronavirus infection remains unproven. Despite an overall amelioration of symptoms, an increase in the number of patients with rhythm disturbances due to the formation of foci of fibrosis after SARS-CoV-2 infection should be expected, especially in patients from the first wave.

## Figures and Tables

**Figure 1 diagnostics-13-01645-f001:**
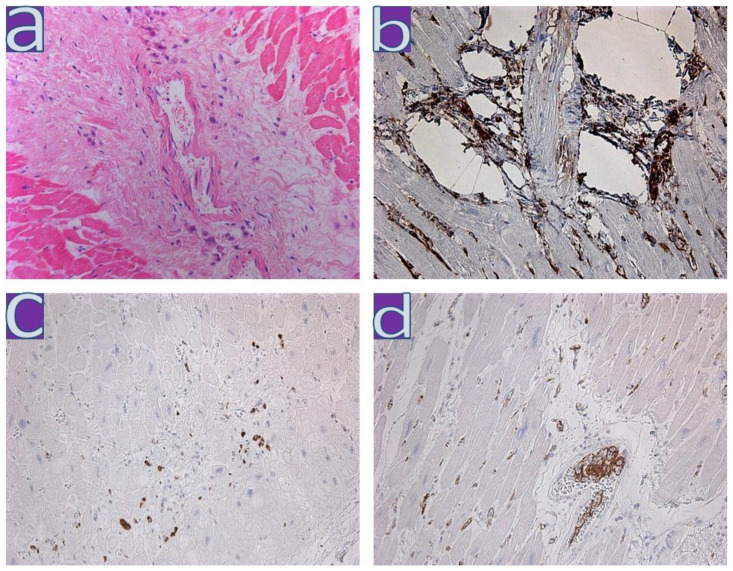
Large polymorphic cells in the myocardium of patient with coronavirus infection. (**a**)—perivascular large polymorphic cells with nuclear hypertrophy; H&E; (**b**)—CD34 expression in large polymorphic cells, (**c**)—CD68 expression in large polymorphic perivascular cells, (**d**)—VEGF expression in large polymorphic, activated endotheliocytes; ×200.

**Figure 2 diagnostics-13-01645-f002:**
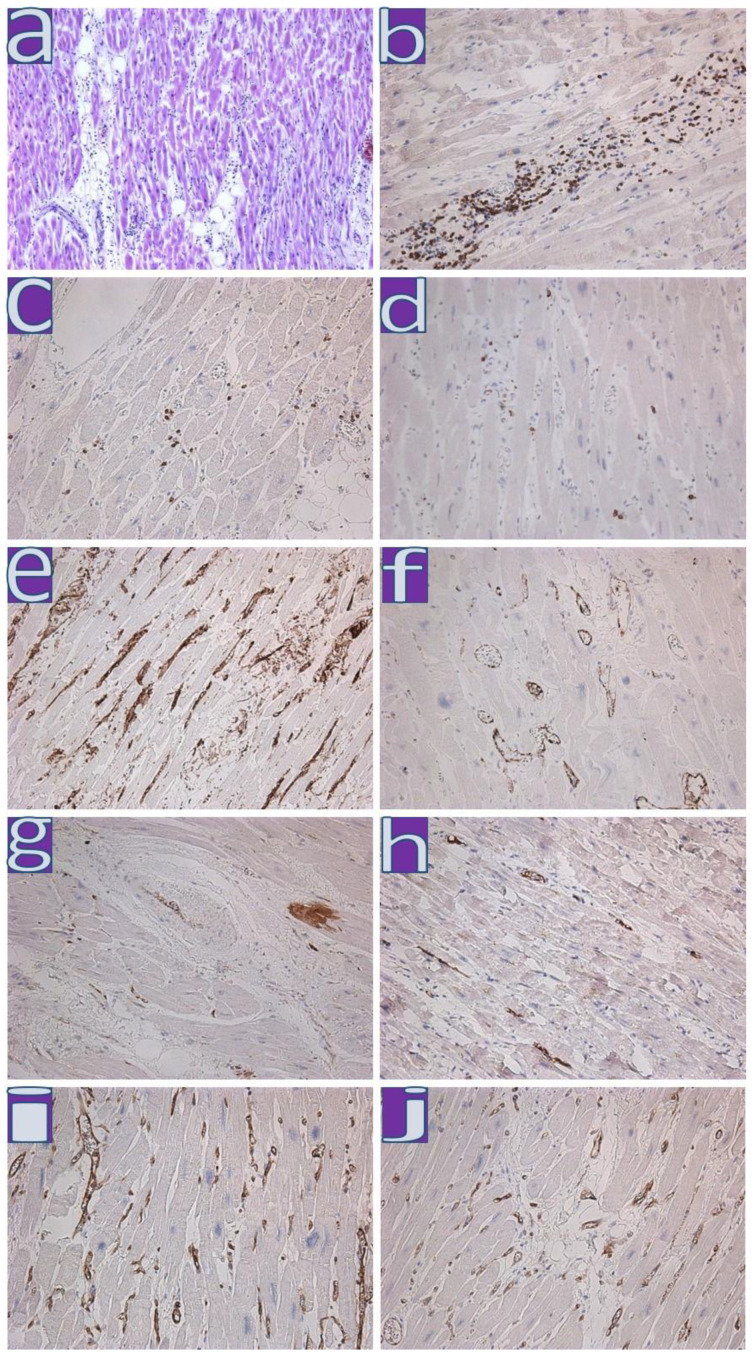
Immunohistochemical study of the myocardium of patients with active myocarditis. (**a**)—active lymphocytic myocarditis; H&E, ×100; (**b**)—CD45 expression on lymphocytes of a large infiltrate in the myocardium; (**c**,**d**)—expression of CD3, CD8 on infiltrate lymphocytes (respectively); (**e**–**h**)—expression of CD34, VEGF, Ang2, and VWF (respectively) on activated endotheliocytes and Ang2 in cardiomyocytes; (**i**,**j**)—expression of MHC1 and HLA-DR (respectively) on vessels and infiltrate cells; ×200.

**Figure 3 diagnostics-13-01645-f003:**
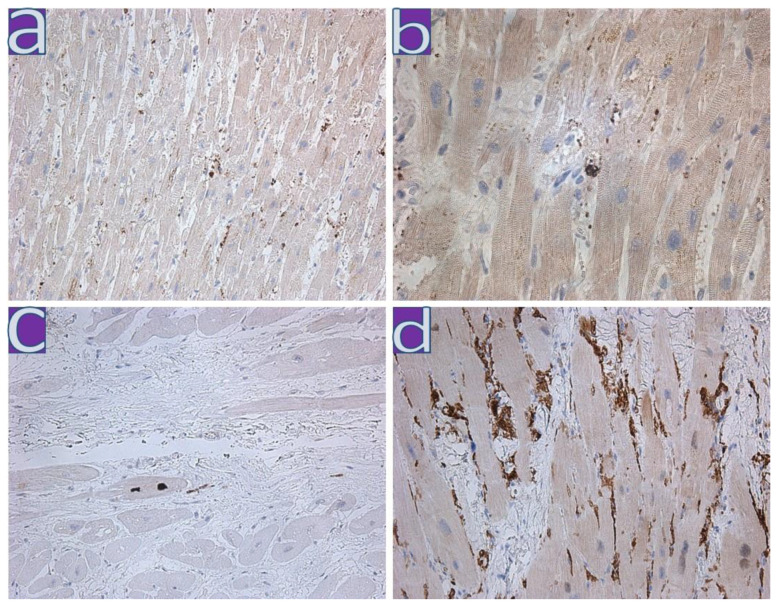
Immunohistochemical study of the myocardium of patients with myocarditis. (**a**–**c**)—SARS-CoV-2 Spike protein expression in vascular endothelium, macrophages, and rare cardiomyocytes; (**d**)—VP1 expression of enteroviruses in myocardial vessels; (**a**)—×100, (**b**–**d**)—×200.

**Figure 4 diagnostics-13-01645-f004:**
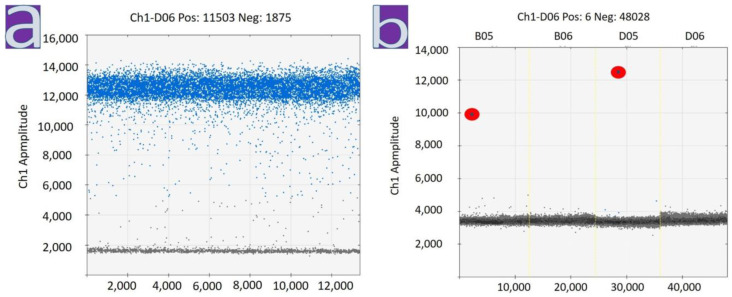
RT-ddPCR analysis of RNA. Single non-reproducible SARS-CoV-2-positive droplets have been identified using FFPE samples of myocardial tissue from patients with myocarditis (marked with a red oval). (**a**) Results of ddPCR for Act-B transcript, isolated from FFPE samples. Multiple positive droplets confirm suitability of RNA for ddPCR analysis; (**b**) Results of ddPCR for Sars-Cov-2 specific RNA (gene N). Single non-reproducible positive droplets have been identified (marked with a red oval).

**Figure 5 diagnostics-13-01645-f005:**
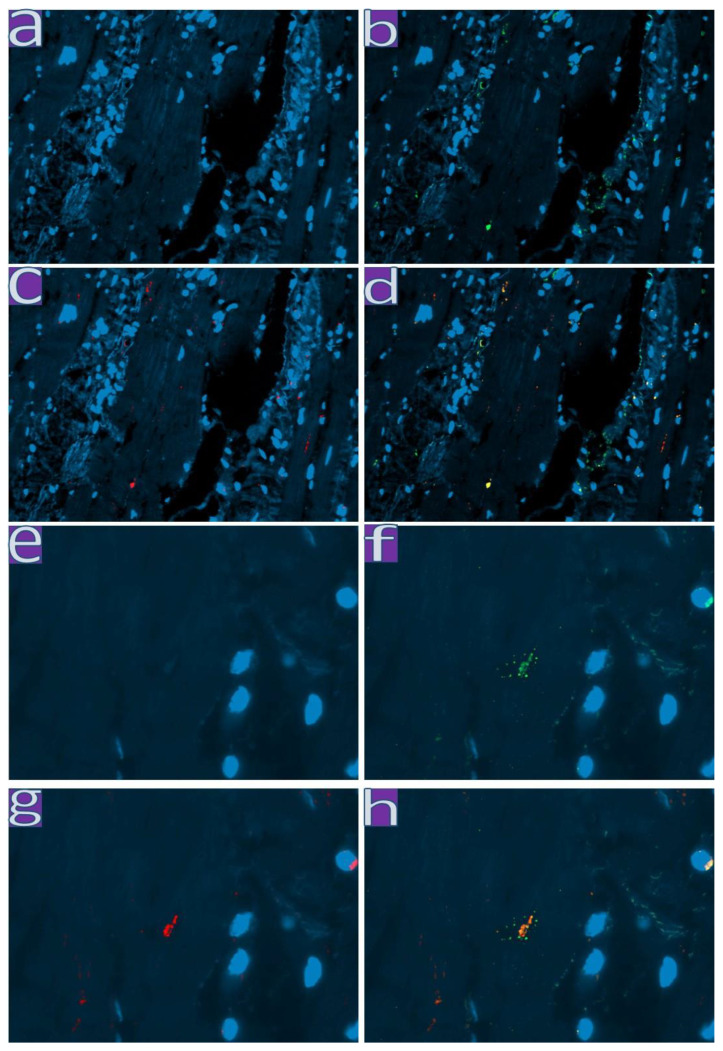
Immunofluorescent study of the myocardium with a cocktail of antibodies against SARS-CoV-2 Spike protein/enterovirus VP1 protein. Green represents fluorescence of CoV-2 Spike protein, red represents fluorescence of enterovirus VP1, yellow or orange represent fluorescence of colocalization of viral antigens. Nuclei were counterstained with DAPI. (**a**–**d**)—×200. (**e**–**h**)—×630.

**Figure 6 diagnostics-13-01645-f006:**
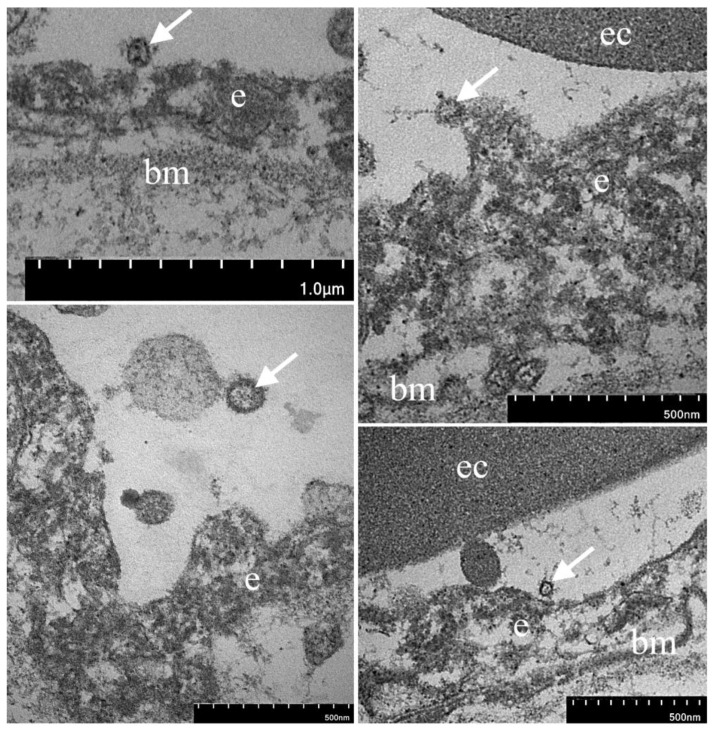
Electron microscopy examination of myocardial vessels. On the apical surface of endotheliocytes, membrane particles morphologically similar to SARS-CoV-2 virions are found (arrows). Legends: e—endotheliocytes, bm—basal membrane, ec—erythrocytes.

**Figure 7 diagnostics-13-01645-f007:**
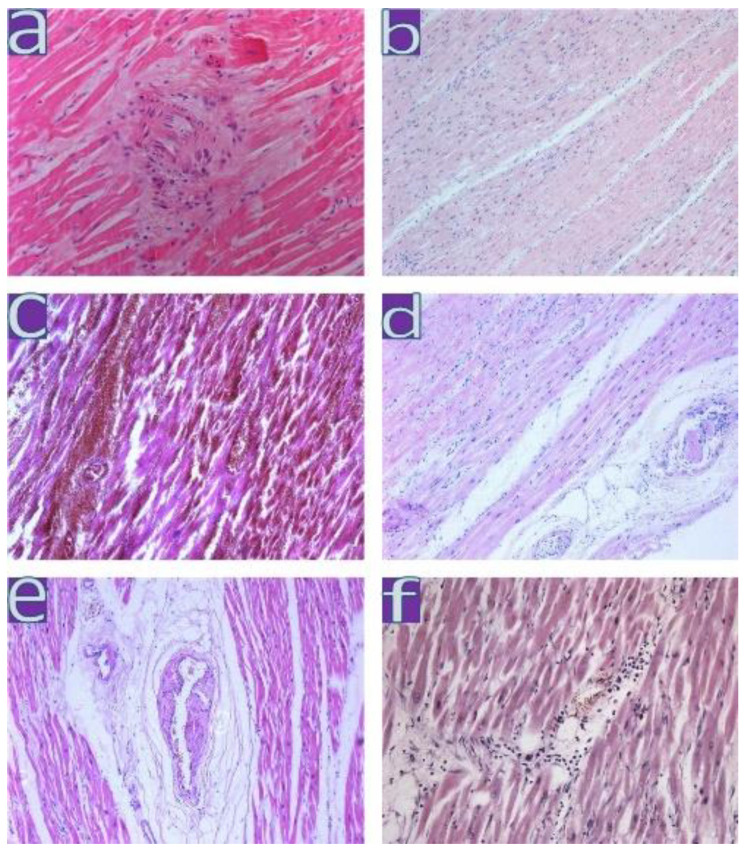
Changes in the myocardium during coronavirus infection. (**a**)—large polymorphic cells in the myocardium; ×200; (**b**)—non-coronary ischemic necrosis; ×100, (**c**)—hemorrhagic myocardial infarction; ×100; (**d**)—thrombi in intramural arteries, myocardial necrosis; ×100; (**e**)—focal plasma impregnation of the intima of the intramural arteries; ×100; (**f**)—active lymphocytic myocarditis; ×100; H&E.

**Figure 8 diagnostics-13-01645-f008:**
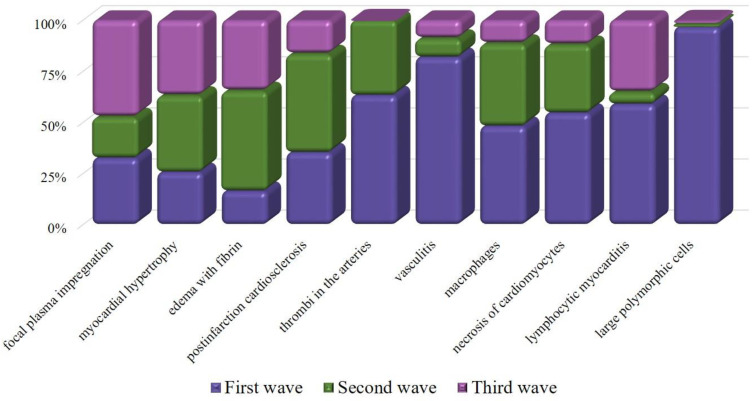
Columns of different colors demonstrate the severity of the feature in the wave.

**Table 1 diagnostics-13-01645-t001:** Primers and probes sequences.

Assay Designation	Genbank Accession	Name	Sequence (5′ to 3′)
E	NC_045512.2	E-F1	ACAGGTACGTTAATAGTTAATAGCGT
		E-R2	ATATTGCAGCAGTACGCACACA
		E-P1	FAM-ACACTAGCCATCCTTACTGCGCTTCG-RTQ1
N	NC_045512.2	N1	GGGGAACTTCTCCTGCTAGAAT
		N2	CAGACATTTTGCTCTCAAGCTG
		N3	FAM-TTGCTGCTGCTTGACAGATT-RTQ1
Act	NM_001101.5	Act-F	ACCGAGCGCGGCTACAG
		Act-R	GGCCATCTCTTGCTCGAAGT
		Act-Pr	FAM-AGCGGGAAATCGTGCGTGACATTA-RTQ1

**Table 2 diagnostics-13-01645-t002:** Clinical and laboratory characteristics of patients with SARS-CoV-2 infection.

	Patients without Myocarditis			Patients with Myocarditis		
	I Wave (*n* = 77)	II Wave (*n* = 119)	III Wave (*n* = 78)	I Wave (*n* = 8)	II Wave (*n* = 1)	III Wave (*n* = 5)
Age (year)	74 (IQR 66–79)	74 (IQR 71–80)	72 (IQR 68–75)	62 (IQR 61–68)	58	82 (IQR 73–84)
Time for onset of symptoms (day)	5 (IQR 3–6)	7 (IQR 6–7)	7 (IQR 8–9)	4 (IQR 4–10)	7	8 (IQR 7–11)
Time from onset of symptoms to death (day)	18 (IQR 14–20)	19 (IQR 17–22)	20 (IQR 16–24)	23 (IQR 20–23)	29	22 (IQR 16–23)
C-reactive protein (mg/L)	112.9 (IQR 60.6–114.1)	88.3 (IQR 64.3–105.1)	87.3 (IQR 54.1–116.9)	124.5 (IQR 161.0–250.0)	113.1	34.7 (IQR 27.8–50.5)
D-dimer (mcg/mL)	2.1 (IQR 1.5–2.7)	0.7 (IQR 0.6–0.8)	0.6 (IQR 0.5–0.7)	1.5 (IQR 1.3–4.9)	1.0	0.4 (IQR 0.2–0.4)
Troponin (ng/mL)	200 (IQR 130–230)	100 (IQR 100–100)	100 (IQR 100–100)	570 (IQR 120–390)	100	100 (IQR 100–670)
Ferritin (mcg/L)	1159 (IQR 846–1880)	621 (IQR 461–735)	869 (IQR 638–1121)	1246 (IQR 420–3175)	1124	341 (IQR 420–1233)
Procalcitonin (ng/mL)	0.49 (IQR 0.45–1.84)	0.15 (IQR 0.11–0.18)	0.14 (IQR 0.12–0.20)	0.44 (IQR 0.35–0.94)	0	0.15 (IQR 0.03–0.62)
Lymphocytes (×10^9^/L)	0.63 (IQR 0.55–0.80)	0.91 (IQR 0.77–1.04)	0.70 (IQR 0.58–0.76)	0.59 (IQR 0.56–1.58)	0.48	0.72 (IQR 0.61–0.83)
Lung lesions on CT till the moment of hospitalization (%)	75 (IQR 60–80)	45 (IQR 40–56)	40 (IQR 35–50)	83 (IQR 70–85)	75	56 (IQR 20–56)
SpO2 (%)	92 (IQR 88–94)	90 (IQR 89–92)	90 (IQR 88–92)	89 (IQR 85–91)	80	87 (IQR 75–93)
Mean left ventricular ejection fraction till the moment of hospitalization (%)	60 (IQR 55–64)	57 (IQR 53–60)	59 (IQR 47–60)	47 (IQR 30–70)	65	55 (IQR 48–70)

**Table 3 diagnostics-13-01645-t003:** Pathomorphological characteristics of patients with immunohistochemical diagnosis of viral myocarditis.

№	Wave	Changes in the Myocardium	CD3/1mm^2^	CD8/1mm^2^	CD68/1mm^2^	MHC1 (in Points)	HLA-DR (in Points)	C1q (in Points)	VP1 Enterovirus (in Points)	SARS-CoV-2 (in Points)	VEGF (in Points)	Ang1 (in Points)	VWF (in Points)
1	I	Lymphocytic myocarditis	15	0	1	4 V 0 C	2 V 0 C	0 V 0 C	0 V 0 C	0 V 0 C	0 V 0 C	2 V 4 C	3 V 0 C
2	I	Lymphocytic myocarditis	20	0	8	4 V 0 C	2 V 0 C	2 V 0 C	4 V 2 C	0 V 0 C	4 V 0 C	4 V 4 C	3 V 0 C
3	I	Lymphocytic myocarditis Non-coronary necrosis, thrombi in intramyocardial arteries	11	0	6	3 V 0 C	0 V 0 C	1 V 0 C	0 V 0 C	0 V 0 C	4 V 2 C	4 V 4 C	1 V 0 C
4	I	Lymphocytic myocarditis	8	0	14	4 V 0 C	1 V 30 C	2 V 0 C	4 V 0 C	0 V 0 C	4 V 0 C	4 V 2 C	4 V 0 C
5	I	Lymphocytic myocarditis Acute myocardial infarction	15	0	32	3 V 0 C	0 V 0 C	2 V 0 C	4 V 0 C	0 V 0 C	2 V 2 C	4 V 3 C	4 V 0 C
6	I	Lymphocytic myocarditis Non-coronary necrosis, thrombi in intramyocardial arteries	37	0	2	4 V 0 C	3 V 0 C	3 V 0 C	4 V 3 C	0 V 0 C	4 V 2 C	2 V 4 C	1 V 0 C
7	I	Lymphocytic myocarditis	21	0	0	4 V 0 C	4 V 0 C	0 V 0 C	4 V 4 C	1 V 0 C	3 V 1 C	4 V 4 C	1 V 0 C
8	I	Lymphocytic myocarditis	16	0	20	4 V 0 C	0 V 0 C	3 V 0 C	4 V 0 C	0 V 30 C	3 V 0 C	4 V 4 C	4 V 0 C
9	II	Focal lymphocytic myocarditis	123	0	38	4 V 0 C	0 V 0 C	3 V 0 C	4 V 4 C	0 V 0 C	3 V 0 C	4 V 4 C	3 V 0 C
10	III	Focal lymphocytic myocarditis Cardiosclerosis	19	0	56	4 V 0 C	4 V 0 C	2 V 0 C	0 V 4 C	0 V 0 C	1 V 0 C	4 V 4 C	3 V 0 C
11	III	Lymphocytic myocarditis	29	0	33	4 V 0 C	2 V 0 C	0 V 0 C	0 V 2 C	2 V 0 C	1 V 0 C	4 V 4 C	4 V 0 C
12	III	Lymphocytic myocarditis	7	0	25	4 V 0 C	2 V 0 C	0 V 0 C	4 V 0 C	1 V 0 C	4 V 4 C	4 V 4 C	3 V 0 C
13	III	Lymphocytic myocarditis	18	8	30	3 V 0 C	2 V 0 C	2 V 0 C	4 V 0 C	0 V 0 C	4 V 3 C	4 V 4 C	3 V 0 C
14	III	Lymphocytic myocarditis Severe fibrosis of the endocardium	98	25	41	4 V 0 C	4 V 0 C	3 V 0 C	1 V 1 C	0 V 0 C	4 V 4 C	4 V 4 C	4 V 0 C

V—on vascular endothelium; C—on cardiomyocytes; 0—lack of expression; 1—expression on single cells and vessels, 2—on 1/3 of the cells, 3—on 1/2–2/3 of the area of the preparation, 4—on all cells under study.

**Table 5 diagnostics-13-01645-t005:** The severity of the histological features depending on the wave of coronavirus infection.

Histological Features	I Wave *n* = 77	II Wave *n* = 119	III Wave *n* = 78
Large polymorphic cells located in the vascular zone	56%	1%	1%
Infarction/necrosis	18%	11%	4%
Hemorrhagic infarction	6%	6%	1%
Thrombi in the arteries	12%	7%	0%
Myocarditis	19%	1%	6%
Postinfarction cardiosclerosis	19%	26%	9%
Thrombi in the endocardium	0%	0%	0%
Pericarditis	1%	2%	4%
Amyloidosis/glycogenosis	0%	0%	1%
Fragmentation of muscle fibers	29%	23%	36%
Pustular myocarditis	4%	3%	1%
Vasculitis	9%	1%	1%
Macrophages	25%	21%	6%
Hemorrhage	48%	24%	17%
Edema with fibrin	8%	24%	17%
Myocardial hypertrophy	47%	69%	68%
Cancer metastasis	0%	3%	0%
Lipomatosis	3%	10%	6%
Spasm	18%	24%	32%
Microbial emboli	0%	3%	1%
Focal plasma impregnation	13%	8%	19%
Post myocardial cardiosclerosis	0%	3%	3%

## Data Availability

Availability of data and materials. All source data are in the Appendix A to the article, if you need clarifications, or need additional information, you can write to the email: doctormakarovia@gmail.com.

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
