# Peer review of "Comparative Study of the Myocardium of Patients from Four COVID-19 Waves"

_diagnostics, 2023, doi:10.3390/diagnostics13091645_

Round 1
Reviewer 1 Report
1. Authors used “According to several publications”, or “other studies”. But only mentioned one reference for each of them. Please refer to several related studies for these kinds of sentences.
Reviewer 2 Report
Despite the fact that the study is interesting and that the authors have invested considerable effort, there are significant shortcomings that may affect the correctness of the conclusions.
The main ambiguities relate to the research methodology.
Namely, the research included 276 patients who died of COVID-19 during four COVID-19 waves: 77 patients in the first, 119 patients in the second and 78 patients in the third wave. Myocardium samples from autopsies were examined histologically and additionally analyzed by PCR. Myocardial immunohistochemistry was performed for 21 samples, immunofluorescent staining for 3 samples, electron microscopic examination only for 2 myocardial samples (both from the fourth wave).
I do not understand:
- according to which criteria the patients were included in the research
- why there are significant differences in the number of included patients from different waves of COVID-19
why are all patients (samples) not subjected to the same diagnostic (immunohistochemical, immunofluorescent, electron microscopic) methods?
In my opinion, these differences are a very likely cause of bias and subsequently incorrect conclusions.
Reviewer 3 Report
Thank you for the opportunity to review this manuscript. In general, the authors carried out autopsy and analysis on 276 patients who died from COVID-19 during four COVID-19 waves and obtained the incidence of different clinical outcomes. The research on COVID-19-related myocarditis is interesting. The authors have attempted provide a systematic clinical and morphological analysis of this topic. I read through the manuscript and have several comments as below:
1. The authors concluded that a decrease in the incidence of endothelitis complications and myocarditis through the waves. I hope the author can explain whether this result is related to the weakening of virulence of the viruses or to vaccination or other potential causes. Actually, we found lots of myocardial injury associated with myocarditis during the Omicron wave. However, this study analyzed a small number of Omicron patients (only two), and this part does not seem to be satisfactory and complete.
2. Myocarditis is an indirect mechanism or possible cause of COVID-19-related myocardial injury. Can the authors provide information on the number of myocardial injuries (eg. increased hs-cTnI) and further analyze the proportion of myocarditis in COVID-19 patients with myocardial injury?
3. The diagnostic criteria of myocarditis are not mentioned in the manuscript. I hope the authors can add this content, because the choice of diagnostic criteria of myocarditis determines the reliability of the study to a certain extent.
4. Line 14 Spelling error. Please change "therd" to "third".
5. Line 205-207 Why did not use the same PCR method and kit to detect SARS-CoV-2 in the fourth wave?
6. Line 223-224 Please explain why only the samples from the fourth wave of patients were examined by electron microscope. In other words, is there any special significance for electron microscopic examination of myocardial samples from the fourth wave instead of the first three waves?
7. Line 236 I want to know why the proportion of patients with ischemic heart disease is as high as 55%, even 72% in the second wave (Line 290)? Is there any evidence of coronary angiography or CTA?
8. Line 250 Please explain why only 75% of patients with myocarditis have an increase in troponin. Did the remaining 25% of patients have other clinical indicators of myocardial injury?
9. As we know, myocarditis is a typical triphasic disease, and the time line of the disease is very important to recognize the infiltration of myocardial immune cells and the pathological diagnosis of myocarditis. The time information in Table 2 is a very good job! But there are also some minor problems in Table 2, please carefully check the data of troponin in patients with myocarditis from II wave in Table 2.
Thanks again for your scholarly efforts in this important field of science.
Reviewer 4 Report
Background Only a few studies have been devoted to the comparison of patients from different waves of COVID-19. In the present work, we aimed to perform a clinical and morphological analysis of the patients deceased from COVID-19 during four COVID-19 waves. Methods The study included 276 patients who died from COVID-19 during four COVID-19 waves: 77 patients in the first, 119 patients in the second and 78 patients in the therd wave. Myocardium samples from autopsies were examined histologically and additionally were analyzed via PCR. Immunohistochemistry of the myocardium was performed for 21 samples using antibodies against CD3, CD45, CD8, CD68, CD34, Ang1, VWF, VEGF, HLA-DR, MHC1, C1q, enteroviral VP1, and SARS-CoV-2-spike protein. Immunofluorescent staining of three myocardial specimens was done using VP1/SARS-CoV-2 antibodies cocktail. RT-ddPCR analysis was performed for 14 RNA samples extracted from paraffin-embedded myocardium. Electron microscopic study of the myocardium was conducted only for two samples from the fourth wave. Findings Active myocarditis was diagnosed in 5% (15/276) of cases. Among them, expression of VP1 was detected in 86% of samples, and SARS-CoV-2-spike protein was found in individual cells in 22%. Immunofluorescence confirmed the colocalization of VP1 and the SARS-CoV-2 Spike proteins. ddPCR did not confidently detect SARS-CoV-2 RNA in the myocardium in any myocarditis cases. At the same time, qPCR of the myocardium sample from wave IV detected the subthreshold signal of SARS-CoV-2, however, myocarditis in this patient was not confirmed. Electron microscopy has demonstrated several single particles similar to SARS-CoV-2 virions on the surface of the endothelium of myocardial vessels. Comparison of cardiovascular complication incidence between three waves showed that the incidence of hemorrhages (48 vs 24 vs 17%), myocardial necrosis (18 vs 11 vs 4%), blood clots in the intramural arteries (12 vs 7 vs 0%), and myocarditis (19 vs 1 vs 6%) decreased over the time, and CD8-T-killers appeared. Immunohistochemistry confirmed the presence of endotheliitis in all 21 studied cases. Сonclusion Comparison of myocardial damage between patients deceased during three COVID-19 waves showed a decrease in the incidence of endotheliitis complications (thrombosis, hemorrhage, necrosis) and myocarditis through the waves. However, the link between myocarditis and SARS-CoV-2 infection remains unproven. Funding This research was financially supported by the Ministry of Science and Higher Education of the Russian Federation (Agreement No. 075-15-2022-301). Can be improved in writing styleAuthor Response
Please see the attachment

Round 2
Reviewer 2 Report
.